# Evaluation of Blood Tumor Mutation Burden for the Efficacy of Second-Line Atezolizumab Treatment in Non-Small Cell Lung Cancer: BUDDY Trial

**DOI:** 10.3390/cells12091246

**Published:** 2023-04-25

**Authors:** Cheol-Kyu Park, Ha Ra Jun, Hyung-Joo Oh, Ji-Young Lee, Hyun-Ju Cho, Young-Chul Kim, Jeong Eun Lee, Seong Hoon Yoon, Chang Min Choi, Jae Cheol Lee, Sung Yong Lee, Shin Yup Lee, Sung-Min Chun, In-Jae Oh

**Affiliations:** 1Department of Internal Medicine, Chonnam National University Hwasun Hospital, Chonnam National University Medical School, Gwangju 58128, Republic of Korea; ckpark214@jnu.ac.kr (C.-K.P.); ohj4250@naver.com (H.-J.O.); repair2799@hanmail.net (H.-J.C.); kyc0923@jnu.ac.kr (Y.-C.K.); 2Department of Medical Science, Asan Medical Institute of Convergence Science and Technology, Asan Medical Center, University of Ulsan College of Medicine, Seoul 05505, Republic of Korea; 3143137@naver.com (H.R.J.); easy0@hanmail.net (J.-Y.L.); 3Department of Internal Medicine, Chungnam National University Hospital, Daejeon 35015, Republic of Korea; jelee0210@cnu.ac.kr; 4Department of Internal Medicine, Pusan National University Yangsan Hospital, Pusan 50612, Republic of Korea; dognose79@naver.com; 5Department of Oncology, Asan Medical Center, University of Ulsan College of Medicine, Seoul 05505, Republic of Korea; ccm9607@gmail.com (C.M.C.); jclee@amc.seoul.kr (J.C.L.); 6Department of Internal Medicine, Korea University Guro Hospital, Seoul 08308, Republic of Korea; syl0801@korea.ac.kr; 7Department of Internal Medicine, School of Medicine, Kyungpook National University, Daegu 41944, Republic of Korea; shinyup@knu.ac.kr; 8Department of Pathology, Asan Medical Center, University of Ulsan College of Medicine, Seoul 05505, Republic of Korea

**Keywords:** cell-free DNA, blood tumor mutation burden, atezolizumab, non-small cell lung cancer

## Abstract

This study aimed to investigate the feasibility of blood-based biomarkers, including blood tumor mutation burden (bTMB), to predict atezolizumab efficacy in relapsed and advanced non-small cell lung cancer (NSCLC). Stage IV NSCLC patients who had previously received platinum-doublet chemotherapy were recruited and received 1200 mg of atezolizumab every three weeks. Blood was collected to obtain plasma cell-free DNA (cfDNA) before the first cycle (C0) and at the fourth cycle (C4). bTMB was measured by CT-ULTRA in patients with cfDNA over 10 ng. The objective response rate (ORR) of the enrolled 100 patients was 10%, and there was no difference in ORR according to bTMB (cutoff: 11.5 muts/Mb) at C0 (high bTMB: 8.1% vs. low bTMB: 11.1%). However, the C4/C0 bTMB ratio was significantly lower in the durable clinical benefit (DCB) patients. The cfDNA concentration at C0, the C4/C0 ratio of the cfDNA concentration, the highest variant allele frequency (hVAF), and the VAF standard deviation (VAFSD) were significantly lower in the DCB patients. In the multivariate analysis, a high cfDNA concentration at C0 (cutoff: 8.6 ng/mL) and a C4/C0 bTMB ratio greater than 1 were significantly associated with progression-free survival. These results suggest that baseline levels and dynamic changes of blood-based biomarkers (bTMB, cfDNA concentration, and VAFSD) could predict atezolizumab efficacy in previously treated NSCLC patients.

## 1. Introduction

The introduction of immune checkpoint inhibitors (ICIs) has improved the overall prognosis of advanced non-small cell lung cancer (NSCLC) in pretreated and treatment-naïve patients [1,2,3,4]. However, the durable antitumor activity of ICI monotherapy has been limited to a subset of patients [5]. Programmed death-ligand 1 (PD-L1) expression with a tumor proportion score (TPS) of at least 50% has been a reliable biomarker for ICI response. However, the predictive power of PD-L1 may be reduced in patients with a TPS of less than 50% [6] or second- or third-line treatment settings in which atezolizumab, an anti-PD-L1 inhibitor, is used regardless of PD-L1 status [4]. Therefore, studies are ongoing to identify additional predictive tumor-based biomarkers for ICI response, such as tumor mutational burden (TMB), microsatellite instability, mismatch-repair deficiency, and gene expression signatures [7,8]. However, other candidates are not currently relevant in practice due to the lack of accessibility or heterogeneity of the outcomes.

A high TMB could predict survival benefits independent of high PD-L1 expression [9]. The measurement of blood TMB has been validated in various studies. In a retrospective analysis performed on POPLAR and OAK study samples, TMB was reproducibly measured in plasma, and the blood TMB (bTMB) was positively correlated with the tissue TMB [10]. Furthermore, a higher bTMB (≥16) was associated with a greater objective response rate (ORR) and increased progression-free survival (PFS) [10].

Besides bTMB, various blood-based biomarkers using circulating tumor DNA (ctDNA) have been discovered in numerous studies. A previous comprehensive analysis using combinations of ctDNA assays, tissue-based biomarkers, peripheral blood, and immune cells was performed to identify patients who were more likely to respond to ICI in advanced NSCLC [11,12,13]. Therefore, this study aimed to investigate the feasibility of plasma DNA-based biomarkers, including bTMB, as potential predictors of atezolizumab efficacy in relapsed or metastatic NSCLC.

## 2. Materials and Methods

### 2.1. Study Design, Patients, and Procedures

This trial was a single-arm, prospective, multi-center cohort study and involved the collection of blood samples from patients receiving atezolizumab. The name ‘BUDDY’ derived from the phrase “evaluation of Blood tumor mUtation burDen for improveD efficacY of atezolizumab”. Between December 2019 and April 2021, patients with locally advanced or metastatic (IIIB-IV or relapsed) NSCLC were recruited from six tertiary centers in South Korea. Patients with NSCLC who had previously received one or two chemotherapy regimens, including at least one platinum-based regimen, were enrolled in this study. The inclusion and exclusion criteria are described in Appendix B.

Patients received 1200 mg of atezolizumab intravenously every three weeks. The treatment continued until the patient experienced a serious adverse event, lost clinical benefit, had confirmed disease progression or withdrew informed consent. Computed tomography (CT) was performed to analyze the tumor at the baseline and every three cycles after that. Clinical responses to the treatment were defined according to Response Evaluation Criteria in Solid Tumors (RECIST) version 1.1 [14].

### 2.2. Blood Sample Collection and Processing

Whole blood samples (20 mL) were collected in K2-EDTA tubes (BD, Franklin Lakes, NJ, USA) before the first cycle (C0) and at the fourth cycle (C4). The C4 samples were collected between the tumor response assessment date (after the third cycle) and the start of the fourth cycle. If the treatment was terminated before C4, blood samples were collected at the end of treatment (EOT). The samples were transported to the central laboratory (Asan Medical Center, Seoul, Republic of Korea), plasma cell-free DNA (cfDNA) was acquired, and the bTMB was analyzed.

Blood samples were centrifuged at 1000× *g* for ten minutes to obtain plasma. Next, the plasma was transferred to a new 2 mL centrifuge tube and subjected to centrifugation at 16,000× *g* and 4 °C for ten minutes to remove cellular debris. The clear plasma was stored in 1 mL aliquots at −80 °C until use. The plasma was isolated within two hours of blood collection to prevent DNA contamination from blood cells. The collection of plasma and the buffy coat were mandatory in enrolled patients. Genomic DNA (gDNA) was extracted from the buffy coat using the QIAamp DNA Blood mini-DNA extraction kit (Qiagen, Hilden, Germany) following the manufacturer’s instructions. The gDNA quantity and purity were measured using a Nanodrop 1000 (Thermo Fisher Scientific, Waltham, MA, USA).

### 2.3. Preparation and Quantification of cfDNA

The cfDNA was isolated from 2.8–6.0 mL of plasma (mean 4.4 mL; maximized plasma volume available) via affinity-based binding to magnetic beads following the manufacturer’s instructions (QIAamp MinElute cfDNA kit, QIAGEN, Hilden, Germany). The cfDNA was eluted in 50 μL of ultraclean water, and the quantity of the cfDNA was measured using a Qubit™ dsDNA HS Assay kit (Thermo Fisher Scientific, Waltham, MA, USA). The quality was measured using a Bioanalyzer 2100 (Agilent Technologies, CA, USA).

### 2.4. Targeted NGS and Analysis of bTMB and Other Biomarkers

Next-generation sequencing (NGS) was performed using CT-ULTRA, a targeted NGS system designed for ctDNA analysis, on patients’ samples with more than 10 ng of cfDNA. The NGS targeted 118 genes, including the entire exons of 88 genes, 30 hotspots and the partial introns of 4 genes often rearranged in cancer. The library preparation for each cfDNA was performed by sequential reactions of end repair, A-base tailing and ligation with a TruSeq adaptor using a SureSelectXT Reagent kit (Agilent Technologies, Santa Clara, CA, USA). After quantification using a Qubit™ (Thermo Fisher Scientific), ten libraries with sample-specific six base pair (bp) barcodes were pooled for a total of 1500 ng to be used for the hybrid capture using an Agilent SureSelectXT custom kit (OP_AMC_ctV1.3 RNA bait; Agilent Technologies). The DNA libraries that passed quality checks were sequenced using the HiSeq X Ten platform (Illumina, San Diego, CA, USA) for paired-end sequencing. Sequenced reads were aligned to the human reference genome (NCBI build 37) with Burrows–Wheeler Aligner (version 0.5.9, broadinstitute, Boston, MA, USA) using the default options. A polymerase chain reaction (PCR) de-duplication was performed using Picard’s MarkDuplicates package. After the initial alignment process, the reads were realigned at common indel positions with the Genome Analysis Toolkit (GATK) IndelRealigner. Then the recalibration of the base quality was done using the GATK Table-Recalibration and used as the final Binary Alignment Map (BAM).

### 2.5. Variant Calling and Filtering for Analysis of the bTMB and Other Candidate Biomarkers

Somatic variant calling of single-nucleotide variants (SNVs) and short insertions and deletions (InDels) for each ctDNA were conducted with the matched buffy coat DNA, using Mutect (version 1.1.6) and the SomaticIndelocator tool in the GATK. Common variants from the somatic variant candidates were filtered out with the common dbSNP (build 141; found in >1% of samples), Exome Aggregation Consortium (ExAC; r0.3.1, threshold frequency 0.001), Korean Reference Genome database (KRGDB) and an in-house panel of normal variants. Final somatic variants were annotated using the Variant Effect Predictor (version 79, accessed on 1 September 2021) and then converted to a maf file using vcf2maf (https://github.com/mskcc/vcf2maf, accessed on accessed on 1 September 2021). To analyze the structural variations, the copy number variations (CNVs) and rearrangements were evaluated using the CNVkit and BreaKmer algorithms, respectively. False-positive variants were filtered out with an in-house panel of normal variants and a manual review. The somatic alterations that remained were further analyzed. The bTMB was measured by calculating the number of somatic, whole exonic (synonymous and non-synonymous) and SNVs and InDels mutations per megabase (Mb) of the genome examined. For the somatic alterations with different variant allele frequencies (VAF) detected in each ctDNA sample, the VAF of the alteration with the highest value and the standard deviation of the VAFs for the alteration were defined as hVAF and VAFSD for each ctDNA, respectively.

### 2.6. Outcomes and Statistical Analyses

The primary endpoint was the ORR in the bTMB-high (bTMBhi) and bTMB-low (bTMBlo) populations. The two groups were divided by the median value of the bTMB at C0 in the ctDNA biomarker-evaluable population (BEP). This trial was an exploratory biomarker study, and the sample size was not based on a statistical calculation.

The secondary endpoints were PFS, overall survival (OS), duration of response (DoR), and clinical benefit rate (CBR) in the intention-to-treat (ITT) population and subgroups. The assessment of the toxicity profile was performed in the safety-evaluable ITT population (SEP). PFS was defined as the time (in months) from the first day of atezolizumab treatment to the day of objective disease progression, death or withdrawal of consent. OS was measured from the first day of ICI administration to the day of death, last day of follow-up or withdrawal of consent. DoR was defined as the time from the first day of the confirmed response to the atezolizumab treatment to the day of objective disease progression, death or withdrawal of consent among patients who experienced a complete response (CR) or partial response (PR). Durable clinical benefit (DCB), one of the criteria of CBR, was defined as survival without disease progression (CR + PR + stable disease (SD)) at 24 weeks [15]. Adverse events were graded based on severity, using the Common Terminology Criteria for Adverse Events (CTCAE; version 4.03, Bethesda, MD, USA).

Data on PD-L1 TPS, EGFR, ALK and ROS1 alterations were collected based on the tests conducted by each institution. The derived neutrophil-to-lymphocyte ratio (NLR) (dNLR) was calculated as follows: dNLR = absolute neutrophil count [ANC]/(white blood cell count—ANC) [16]. The subgroups, according to the NLR, dNLR and platelet-to-lymphocyte ratio (PLR), were divided by their median values at C0. The cut-off values of other candidate biomarkers at C0 were calculated based on the analysis of receiver operating characteristic (ROC) curves and the area under the curve (AUC).

All data were expressed as the mean ± standard deviation (SD) or as the median (range) for the continuous variables and as the number and percentage for the categorical variables. Intergroup comparisons of responses were performed using a Student’s *t*-test or a Mann–Whitney U test for the continuous variables and Pearson’s χ^2^ test or Fisher’s exact test for the categorical variables. Survival times were estimated for each group using the Kaplan–Meier method. Univariate and multivariate analyses of survival were performed using the Cox proportional hazards model. Statistical analysis was performed using IBM^®^ SPSS^®^ statistics version 25 (IBM Corp., Armonk, NY, USA) and R statistics [17]. A *p*-value of less than 0.05 was considered significant.

## 3. Results

### 3.1. Patient Characteristics and Efficacy of Atezolizumab

A total of 100 patients were enrolled in this study, and 100 plasma and 100 buffy coat samples were obtained at C0 (Figure 1). The baseline characteristics of the patients are summarized in Table 1 and Appendix A. The comparisons of the characteristics according to the best response and clinical benefit rates of the atezolizumab treatment are described in Appendix A. EGFR mutations were detected in 13.8% of the tested patients. ALK fusion was only detected in one patient (1.4%). PD-L1 immunohistochemistry (IHC) was performed on 95 patients, and high PD-L1 expression (TPS ≥ 50% by 22C3 or SP263 antibody, TC ≥ 50% or IC ≥ 10% by SP142 antibody) was present in 35 patients (36.8%).

The ORR and DCB rates were 10 and 25%, respectively (Table 2). Patients with a PR were associated with higher proportions of a PD-L1 TPS over 50% or higher PD-L1 expression than patients without PR (Appendix A). There were no differences in the ORR and DCB rates according to the NLR, dNLR or PLR at C0 (Appendix A).

At the median follow-up of 12.3 months, the median PFS and OS of the ITT population were 2.1 months (95% confidence interval (CI): 1.6–3.0) and 13.1 months (95% CI: 10.1–16.2), respectively. Patients with high PD-L1 expression had a longer PFS compared with patients with a low-to-negative PD-L1 expression (*p* = 0.019; Appendix A). Patients with a low NLR (<2.84) at C0 had better OS benefits compared with those with a high NLR (≥2.84) at C0 (Appendix A). At the time of the data cut-off (29 October 2021), progression was confirmed in 76 patients, and 46 had received subsequent chemotherapy (Appendix A).

Adverse events developed in 82 patients (Appendix A). Treatment interruption and withdrawal were reported in 14 and 23 patients, respectively. Suspicious immune-related adverse events (irAEs) developed in 41 patients; irAEs with a grade of 3 or more were reported in two patients. Except for a simple elevation in the liver-function test, skin manifestations were the most common irAEs.

### 3.2. bTMB as a Biomarker for Atezolizumab Response

Among the 100 cfDNA samples, cfDNA concentration was measured in 86 samples; the samples that failed quality control (*n* = 11) and non-evaluable group samples (*n* = 3) were excluded. Among the 86 cfDNA samples, the bTMB was measured in 64 ctDNA-positive samples at C0 and 48 paired samples at C4/EOT (Figure 1).

The median value of bTMB in the ctDNA BEP (*n* = 64) was 11.5 Mut/Mb (Figure 2a). There was no correlation between the bTMB and cfDNA concentration (Figure 2b). There was no difference in the mean level of bTMB at C0 between the DCB and NDB groups (Figure 2c). However, the mean level of bTMB decreased from C0 to C4/EOT in the DCB group but increased in the NDB group (Figure 2d). Furthermore, the C4 to C0 bTMB ratio was significantly different between the DCB and NDB groups (Figure 2e). When divided by the median value of bTMB at C0 (11.5 Mut/Mb), there was no difference in the ORR (bTMBhi 8.1% vs. bTMBlo 11.1%; *p* = 1.000) and the DCB rate (bTMBhi 18.9% vs. bTMBlo 18.5%; *p* = 1.000) (Figure 2f). The ORR and DCB rates were higher in the subgroup with no change or decreased bTMB levels from C0 to C4/EOT than in the subgroup with increased bTMB levels although the difference was not statistically significant (Figure 2g).

There was no difference in the PFS according to the bTMB level at C0; however, the subgroup with no change or decreased bTMB levels from C0 to C4/EOT had a longer PFS compared with the subgroup with increased bTMB levels (Figure 2h). There was no significant difference in the OS between the subgroups according to the bTMB level at C0 or the dynamic change in the bTMB level from C0 to C4/EOT (Figure 2i).

### 3.3. cfDNA Concentration as a Biomarker for Atezolizumab Response

The cfDNA concentration of 86 cfDNA samples at C0 and 64 paired C4/EOT samples was measured (Figure 1). The mean value of cfDNA concentration at C0 was significantly lower in the DCB group than in the NDB group (Figure 3a). Based on that result, the optimal cut-off value for DCB was determined to be 8.6 ng/mL (AUC 0.699, *p* < 0.001) (Figure 3b). The mean level of cfDNA concentration from C0 to C4/EOT increased in the NDB group (Figure 3c). Furthermore, the C4/C0 ratio of the cfDNA concentration was significantly higher in the NDB group than the DCB group (Figure 3d).

At the cut-off value at C0 (8.6 ng/mL), patients with a low cfDNA concentration at C0 had significantly higher DCB rates (32.7% vs. 9.3%; *p* = 0.005) (Figure 3e). Patients with no change or a decreased cfDNA concentration level from C0 to C4/EOT had higher DCB rates compared with those with an increased cfDNA concentration level (40.9% vs. 21.4%; *p* = 0.176) (Figure 3f).

The subgroup with low cfDNA concentrations at C0 had a longer PFS and OS compared with the subgroup with high cfDNA concentrations; however, there was no difference in the PFS and OS according to the dynamic change in the cfDNA concentration from C0 to C4/EOT (Figure 3g–h).

### 3.4. ctDNA hVAF as a Biomarker for Atezolizumab Response

The ctDNA hVAF of 64 ctDNA samples at C0 and 48 paired samples at C4/EOT was measured (Figure 1). The mean value of hVAF at C0 was significantly lower in the DCB group than in the NDB group (Figure 4a), and the difference in hVAF was consistent regardless of the magnitude of bTMB (Figure 4b). Based on that result, the optimal cut-off value for DCB was determined to be 3.9% (AUC 0.579, *p* < 0.001) (Figure 4c). The mean level of hVAF from C0 to C4/EOT decreased in the DCB group and increased in the NDB group (Figure 4d). In addition, the C4/C0 hVAF ratio was significantly higher in the NDB group than in the DCB group (Figure 4e).

At the cut-off value at C0 (3.9%), there was no difference in the DCB rate (high 13.8% vs. low 22.9%; *p* = 0.546) according to the hVAF level at C0 (Figure 4f). However, patients with no change or a decreased hVAF from C0 to C4/EOT had higher DCB rates compared with those with an increased hVAF (50.0% vs. 9.4%; *p* = 0.005) (Figure 4g).

There was no difference in the PFS according to the hVAF level at C0; however, the subgroup with no change or a decreased hVAF level from C0 to C4/EOT had a longer PFS compared with the subgroup with an increased hVAF level (Figure 4h). As opposed to the PFS, the subgroup with a low hVAF at C0 had a longer OS compared with the subgroup with a high hVAF; however, there was no difference in OS between the subgroups according to the dynamic change in the hVAF from C0 to C4/EOT (Figure 4i).

### 3.5. VAFSD in ctDNA as a Biomarker for Atezolizumab Response

The ctDNA VAFSD of 53 ctDNA samples at C0 with two or more detected somatic mutations (bTMB ≥ 7.7 Mut/Mb) and 37 paired samples at C4/EOT was measured (Figure 1). The mean value of VAFSD at C0 was significantly lower in the DCB group than in the NDB group (Figure 5a), and the difference in VAFSD was consistent regardless of the magnitude of the bTMB (Figure 5b). Based on that result, the optimal cut-off value for DCB was determined to be 0.014 (AUC 0.598, *p* < 0.001) (Figure 5c). The mean level of VAFSD significantly increased from C0 to C4/EOT in the NDB group (Figure 5d). Furthermore, the C4/C0 VAFSD ratio was significantly higher in the NDB group than in the DCB group (Figure 5e).

At the cut-off value at C0 (0.014), there was no difference in the DCB rate (high 12.0% vs. low 21.4%; *p* = 0.585) according to the VAFSD level at C0 (Figure 5f). However, patients with no change or a decreased VAFSD from C0 to C4/EOT had a higher DCB rate compared with those with an increased VAFSD level (42.9% vs. 8.7%; *p* = 0.042) (Figure 5g).

There was no difference in the PFS according to the VAFSD level at C0; however, the subgroup with no change or a decreased VAFSD level from C0 to C4/EOT had a longer PFS compared with the subgroup with an increased VAFSD level (Figure 5h). As opposed to the PFS, the subgroup with a low VAFSD at C0 had a longer OS compared with the subgroup with a high VAFSD; however, there was no difference in the OS between the subgroups according to the dynamic change in VAFSD from C0 to C4/EOT (Figure 5i).

### 3.6. Mutation Profiling

In the oncoplot of the analysis of C0 samples, a TP53 mutation was the most common variant (66%), and five of the six patients who had rapid progression in the NDB group had a TP53 mutation. KRAS, NF1, PIK3CA, JAK2, NFE2L2, and RB1 mutations were more frequently detected in the DCB group, and CDKN2A, ARID1A, and EGFR mutations were more common in the NDB group; however, the differences were not statistically significant (Figure 6a).

In the sunburst charts of the analysis of C0 and matched C4/EOT samples, 74 and 269 variants were detected in the DCB and NDB groups, respectively. The proportions of variants only detected at C0 were significantly higher in the DCB group (40.5%) than in the NDB group (7.4%). The proportions of variants only detected at C4/EOT were higher in the NDB group (16.7%) than in DCB group (2.7%) (Figure 6b–d).

### 3.7. Subgroup Analysis for PFS and OS in the cfDNA BEP

In the univariate analysis for the PFS, female patients, liver metastasis, EGFR mutation, low or no PD-L1 expression, cfDNA concentration at C0, dynamic change of bTMB, hVAF, and VAFSD were associated with worse outcomes (Appendix A). In the univariate analysis for the OS, squamous histology, cfDNA concentration at C0, hVAF at C0 and VAFSD at C0 were associated with decreased survival (Appendix A).

A multivariate analysis of the PFS revealed that increased bTMB from C0 to C4 (C4/C0 > 1; hazard ratio (HR): 4.95, 95% CI: 1.63–15.07) and a high cfDNA concentration at C0 (HR: 5.16, 95% CI: 1.73–15.37) were significant risk factors (Appendix A). When stratifying the BEP by these two factors, the Kaplan–Meier analyses of the PFS showed a clear difference among the three groups. The group with an increased bTMB from C0 to C4 and a high cfDNA concentration at C0 showed the worst outcomes (HR: 8.98, 95% CI: 2.73–29.56) (Appendix A).

In the multivariate analysis for the OS, a high VAFSD at C0 (HR:2.84, 95% CI: 1.24–6.50) was confirmed as an independent prognostic factor for the OS (Appendix A). When combining the VAFSD with the cfDNA concentration at C0 and stratifying the BEP by these two factors, the group with a high VAFSD and a high cfDNA concentration at C0 was associated with a worse prognosis (HR: 3.67, 95% CI: 1.13–11.88) (Appendix A).

## 4. Discussion

This study was a single-arm, prospective, observational study and a comprehensive analysis of blood samples as predictive biomarkers for atezolizumab response in previously treated patients with advanced NSCLC. In this study, baseline cfDNA concentration and dynamic change in the hVAF and VAFSD were significant predictors of DCB. Furthermore, a high level of cfDNA concentration at the baseline, an increased level of bTMB from the baseline to the first response assessment or EOT and a high VAFSD at the baseline were significant risk factors for decreased PFS and OS.

Pretreatment ctDNA level, especially bTMB, is known as a surrogate marker of tumor load, and the identification of tumor-related mutations using ctDNA can assist in predicting the ICI response in metastatic solid cancers [18,19]. However, the usefulness of bTMB has been controversial due to its reproducibility and ambiguity of thresholds due to the use of different gene panel platforms [19,20]. The analysis of POLAR and OAK cohorts revealed that the high bTMB subgroup failed to differentiate patients with OS benefits, unlike PFS, regardless of the selected thresholds [10]. In the primary analysis of the BFAST cohort C phase 3 trial, the first-line atezolizumab treatment did not improve the investigator-assessed PFS compared with platinum-based chemotherapy in the population with a bTMB cutoff of 16 or over [21]. Furthermore, false negative results may have occurred due to insufficient plasma ctDNA content or the lack of inclusion of altered genes in the targeted panel [22]. In this study, pretreatment ctDNA could be detected and used for the bTMB measurement in only 64 out of the 100 enrolled patients. In addition, in previously treated advanced lung cancer, the baseline ctDNA level could be affected by various conditions such as type of tumor, progression status, proliferative rate, and treatment modality. Therefore, an absolute value of the pretreatment bTMB may have insufficient power to predict effectiveness, at least for second-line ICI treatments.

In contrast, monitoring dynamic changes in ctDNA could be an alternative for enhancing the utility of the bTMB. In a meta-analysis of ten previous studies, early reduction of ctDNA was associated with improved PFS, OS, and ORR in advanced NSCLC patients treated with ICIs; however, the baseline level of ctDNA was irrelevant to the clinical outcomes [11]. In another biomarker analysis of RCT for the efficacy and safety of camrelizumab (anti-PD-L1 inhibitor) plus chemotherapy in advanced squamous NSCLC, patients with a low on-treatment bTMB after two cycles and a decreased bTMB were associated with a better ORR, PFS, and OS in the camrelizumab plus chemotherapy group [23]. In that study, the pretreatment bTMB was not associated with ORR, PFS, and OS in the camrelizumab and placebo plus chemotherapy groups. An early reduction of ctDNA may reflect an early response of tumors to effective treatments and a decrease in the tumor burden. In this study, a bTMB reduction from C0 to C4/EOT was significantly associated with a PFS benefit, but the pretreatment level of bTMB was irrelevant to the survival benefit. Furthermore, the analysis of the hVAF and VAFSD, which may reflect the tumor burden as a potential ctDNA-related biomarker, also showed a significant influence on the PFS. Therefore, a longitudinal assessment of the ctDNA could be a valuable tool for guiding clinical decisions during ICI treatments.

In a recent biomarker study using POPLAR, OAK and other validation cohorts, the maximum somatic allele frequency (MSAF), defined as the maximum allele frequency (AF) of all the somatic mutations identified per sample by NGS, was adopted in the study to adjust the bTMB for MSAF interference [24]. In those cohorts, the MSAF levels were significantly higher in patients with a high as opposed to low bTMB, and a high MSAF was a significant negative prognostic factor for OS. These results suggested that the detection of bTMB was dependent on the ctDNA amount, and the insufficient predictive power of bTMB may have originated from the interference of a high AF. In the B-F1RST trial of validating the bTMB for first-line atezolizumab in stage IIIB-IV NSCLC, patients with an MSAF less than 1% had a higher ORR than those with an MSAF greater than or equal to 1% [25]. In comparisons of the baseline characteristics between the MSAF less than 1% and MSAF greater than or equal to 1% subgroups, a low MSAF (<1%) was associated with favorable prognostic factors, such as younger ages, fewer current smokers, more patients with a PD-L1 positive status, lower number of target lesions, and smaller tumor sizes, which may suggest a lower tumor burden. In another study on durvalumab (anti-PD-L1 inhibitor) that used three cohorts, a higher pretreatment VAF was associated with a poor OS, and on-treatment reductions in the VAF and a lower on-treatment VAF were independently associated with a longer PFS and OS [26]. In this study, the concept of the highest VAF (hVAF) was applied, similar to the MSAF. The baseline hVAF and dynamic change in hVAF were significantly associated with DCB, PFS, and OS. Although the predictive power of the hVAF was diminished in a multivariate analysis for the PFS and OS, the hVAF may be a relevant blood-based biomarker for tumor burden. However, the hVAF needs to be validated in a larger cohort with a paired analysis at pre- and on-treatment time points.

Intratumoral heterogeneity (ITH) is a major cause of anticancer treatment failure, including resistance to ICIs [27,28]. From the perspective of a second-line treatment setting, previous anticancer therapies may increase genetic ITH by creating new subclones with different somatic mutations. Tumors with highly heterogeneous subclonal structures may not produce enough neo-antigens to enhance the antitumor activity of T-cells after the subsequent ICI [27], suggesting that ITH can be an unfavorable prognostic or predictive factor for ICI treatment. However, there are few comprehensive studies on ITH as a predictive biomarker of ICI in advanced NSCLC due to the lack of tumor tissue-based sequencing data from multiple metastatic regions [29,30]. Meanwhile, several recent studies applied potential methods of plasma-based ITH estimation using ctDNA, such as AF heterogeneity (AFH; the ratio of AF of a mutation to MSAF < 10%) [31], and blood-based ITH (bITH; modifying Shannon diversity index formula) [32]. In these studies, the presence of ctDNA-based ITH at the baseline and a dynamic change in increased ctDNA-based ITH were associated with unfavorable outcomes of ICI [31,32]. In this study, the concept of the VAFSD was applied as a surrogate for ctDNA-based ITH. Similar to previous studies, a high VAFSD at the baseline and an increased VAFSD were strong predictors of DCB, PFS, and OS. Therefore, the VAFSD might be an alternative biomarker for ITH and the antitumor response of ICI.

This study had several limitations. First, compared with the baseline, there were limited numbers of paired C4/EOT samples. The insufficient number of on-treatment samples might have affected the predictive power of the dynamic changes in the novel biomarkers. Second, the timing of the on-treatment sampling varied but was mainly from eight to nine weeks, and blood sampling from three to six weeks was not mandatory in this study. Validation of the reproducibility of the biomarker candidates might be necessary for the earlier phase of treatment before the first imaging study. Finally, the mutation profiling in the ctDNA BEP did not reveal any specific mutations related to the efficacy of the ICI treatment. The genetic ITH or compound mutations developed after a previous anticancer treatment may have affected the indistinct distribution of mutations. A comprehensive mutational profiling comparing tumor tissue and plasma is warranted in a larger prospective cohort. Furthermore, larger clinical trials, including different types of tumors and ICIs, are needed to validate the utility and versatility of these blood-based biomarkers.

## 5. Conclusions

In previously treated advanced NSCLC patients, baseline levels and dynamic changes in blood-based biomarkers can predict the efficacy of atezolizumab. ctDNA can provide a snapshot of the overall genomic profiling for advanced NSCLC and overcome the bias from an individual tumor biopsy. These ctDNA-based biomarkers are easily accessible by using validated NGS panels for NSCLC. This comprehensive analysis of blood-based biomarkers could aid in identifying NSCLC patients who would benefit from ICI in the initial phase of treatment. It may also provide an informative insight into the complex tumor heterogeneity and propose futuristic solutions.

## Figures and Tables

**Figure 1 cells-12-01246-f001:**
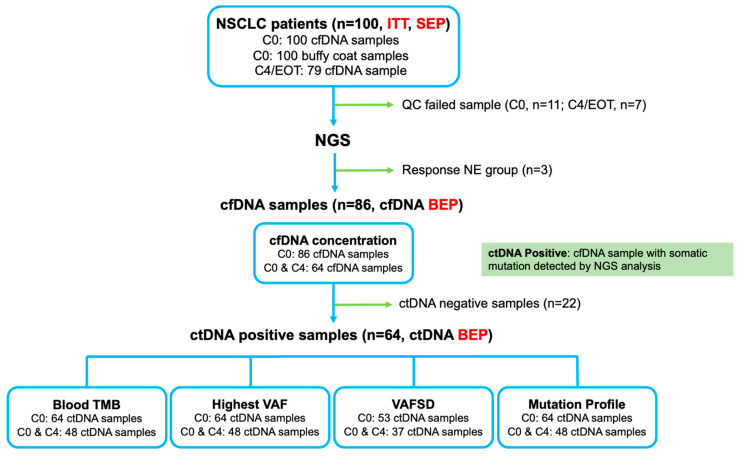
Intention-to-treat (ITT) population and biomarker-evaluable population (BEP). NSCLC, non-small cell lung cancer; SEP, safety-evaluable population; cfDNA, circulating cell-free DNA; QC, quality control; NGS, next-generation sequencing; ctDNA, circulating tumor DNA; TMB, tumor mutation burden; VAF, variant allele frequency; SD, standard deviation.

**Figure 2 cells-12-01246-f002:**
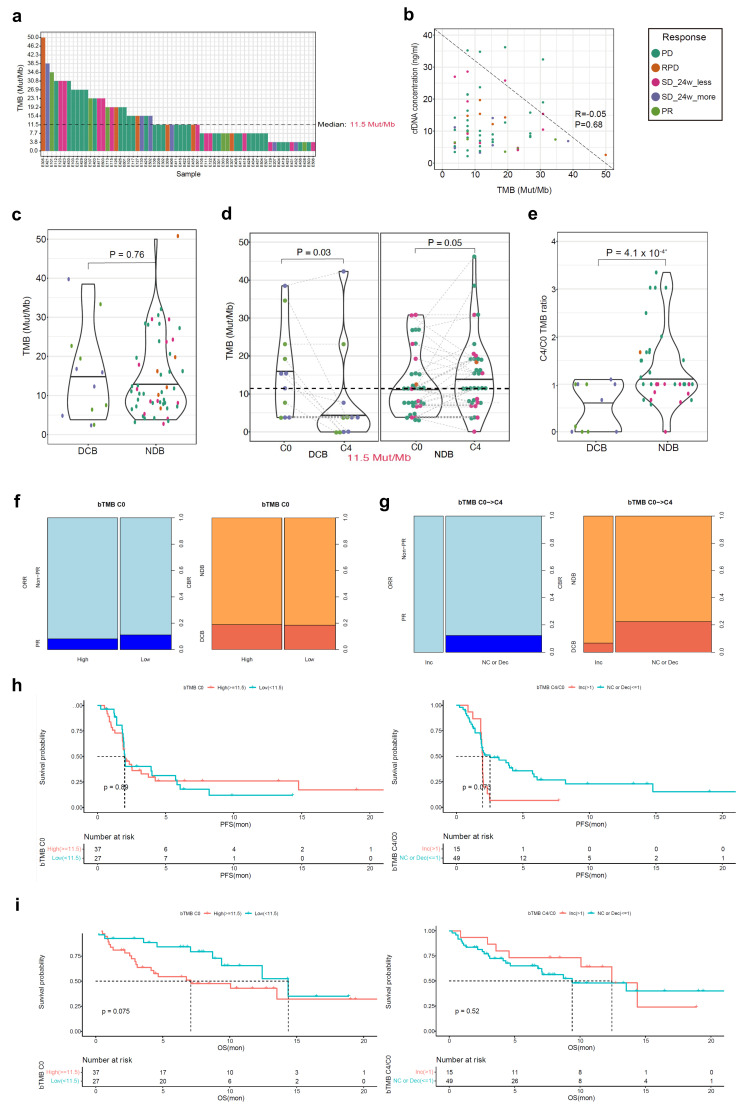
Atezolizumab efficacy according to the bTMB in the ctDNA BEP (*n* = 64). (**a**) Comparison of the bTMB at C0; (**b**) The correlation between the bTMB and cfDNA levels; (**c**) bTMB at C0 according to the CBR; (**d**) The bTMB change between C0 and C4/EOT according to the CBR; (**e**) Comparison of the C4/C0 bTMB ratio according to the CBR; (**f**) Comparison of the ORR and CBR between the C0 bTMB high and low subgroups; (**g**) Comparison of the ORR and CBR between the subgroups according to the bTMB dynamics; (**h**) The PFS according to the C0 bTMB and bTMB dynamics; (**i**) The OS according to the C0 bTMB and bTMB dynamics. bTMB, blood tumor mutation burden; ctDNA, circulating tumor DNA; BEP, biomarker-evaluable population; cfDNA, circulating cell-free DNA; PD, progressive disease; RPD, rapid progressive disease; SD, stable disease; PR, partial response; CRB, clinical benefit ratio; DCB, durable clinical benefit; NDB, non-durable benefit; ORR, objective response rate; Inc, increased; NC, no change; Dec, decreased; PFS, progression-free survival; OS, overall survival. * *p* < 0.05.

**Figure 3 cells-12-01246-f003:**
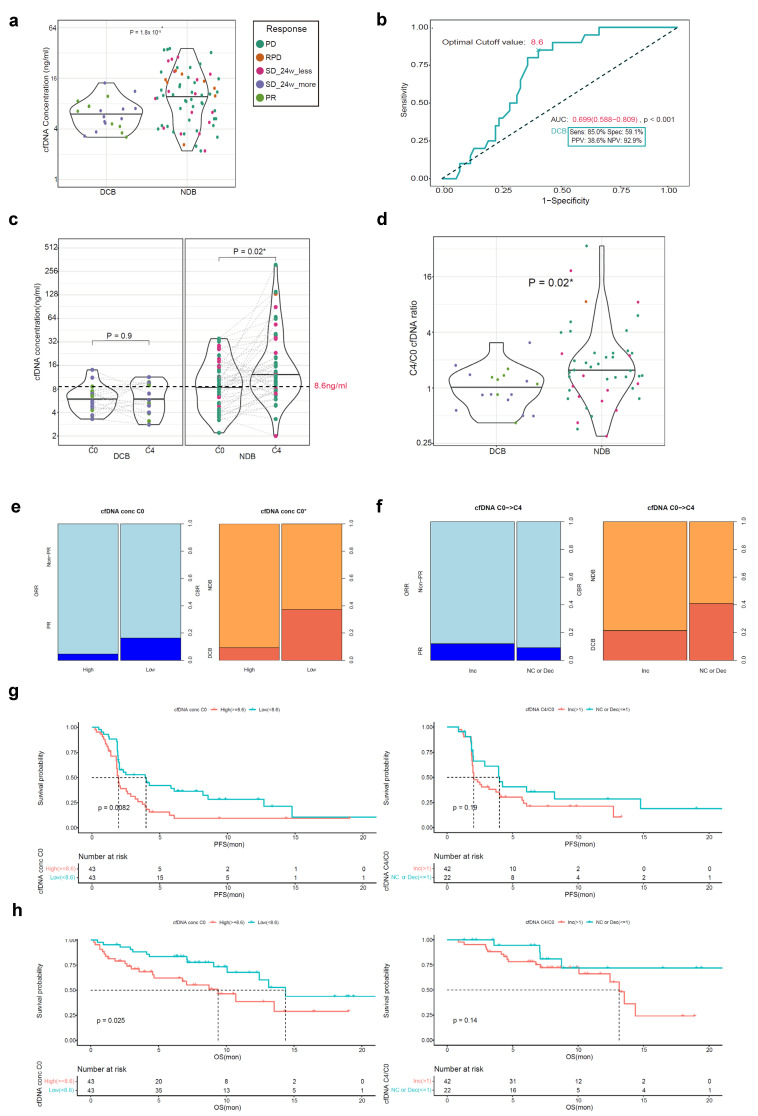
Atezolizumab efficacy according to the cfDNA concentration in the cfDNA BEP (*n* = 86). (**a**) The cfDNA concentration at C0 according to the CBR; (**b**) The ROC curve for the cfDNA concentration at C0; (**c**) The change in cfDNA concentration between C0 and C4/EOT according to the CBR; (**d**) Comparison of the C4/C0 cfDNA concentration ratio according to the CBR; (**e**) Comparison of the ORR and CBR between the C0 cfDNA concentration high and low subgroups; (**f**) Comparison of the ORR and CBR between the subgroups according to the cfDNA concentration dynamics; (**g**) The PFS according to the C0 cfDNA concentration and cfDNA concentration dynamics; (**h**) The OS according to the C0 cfDNA concentration and cfDNA concentration dynamics. * *p* < 0.05. cfDNA, circulating cell-free DNA; BEP, biomarker-evaluable population; CRB, clinical benefit ratio; circulating cell-free DNA; DCB, durable clinical benefit; NDB, non-durable benefit; PD, progressive disease; RPD, rapid progressive disease; SD, stable disease; PR, partial response; ROC, receiver operating characteristic; AUC, area under the curve; PPV, positive predictive value; NPV, negative predictive value; ORR, objective response rate; Inc, increased; NC, no change; Dec, decreased; PFS, progression-free survival; OS, overall survival.

**Figure 4 cells-12-01246-f004:**
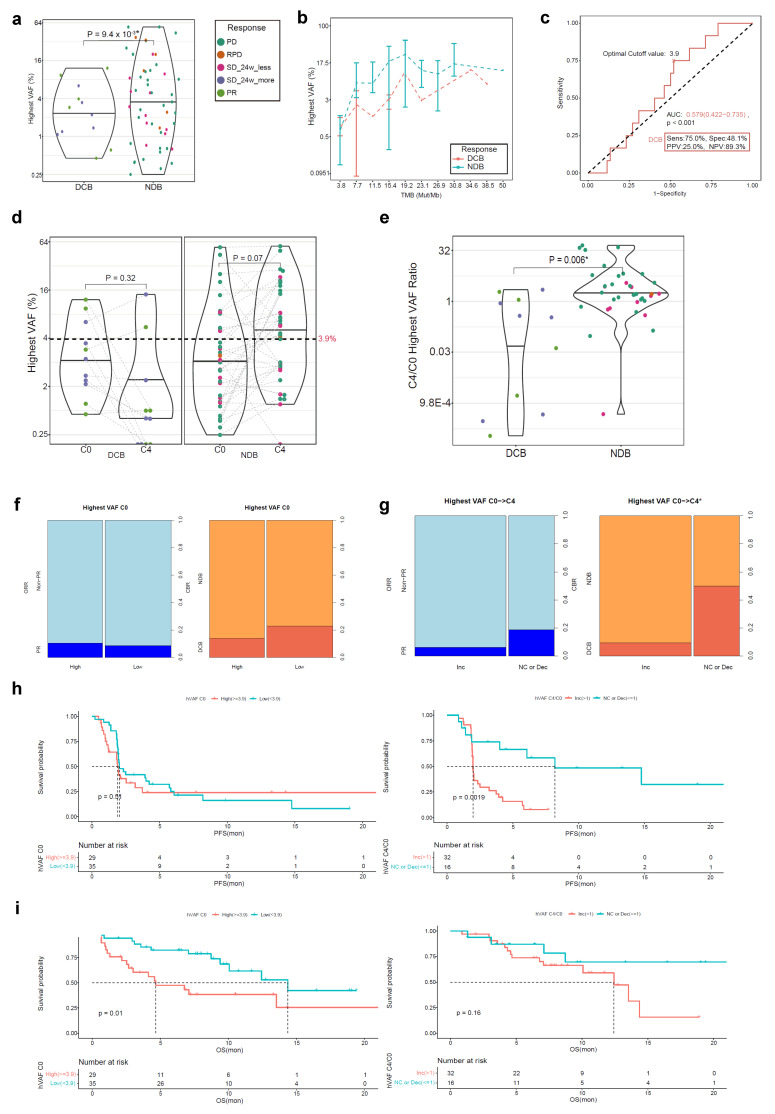
Atezolizumab efficacy according to the hVAF in the ctDNA BEP (*n* = 64). (**a**) The hVAF at C0 according to the CBR; (**b**) Comparison between the hVAF and bTMB at C0; (**c**) The ROC curve for the hVAF at C0; (**d**) The hVAF change between C0 and C4/EOT according to the CBR; (**e**) Comparison of the C4/C0 hVAF ratio according to the CBR; (**f**) Comparison of the ORR and CBR between the C0 hVAF high and low subgroups; (**g**) Comparison of the ORR and CBR between the subgroups according to the hVAF dynamics; (**h**) The PFS according to the C0 hVAF and hVAF dynamics; (**i**) The OS according to the C0 hVAF and hVAF dynamics. * *p* < 0.05. hVAF, highest variant allele frequency; ctDNA, circulating tumor DNA; BEP, biomarker-evaluable population; CRB, clinical benefit ratio; circulating cell-free DNA; DCB, durable clinical benefit; NDB, non-durable benefit; PD, progressive disease; RPD, rapid progressive disease; SD, stable disease; PR, partial response; bTMB, blood tumor mutation burden; ROC, receiver operating characteristic; AUC, area under curve; PPV, positive predictive value; NPV, negative predictive value; ORR, objective response rate; Inc, increased; NC, no change; Dec, decreased; PFS, progression-free survival; OS, overall survival.

**Figure 5 cells-12-01246-f005:**
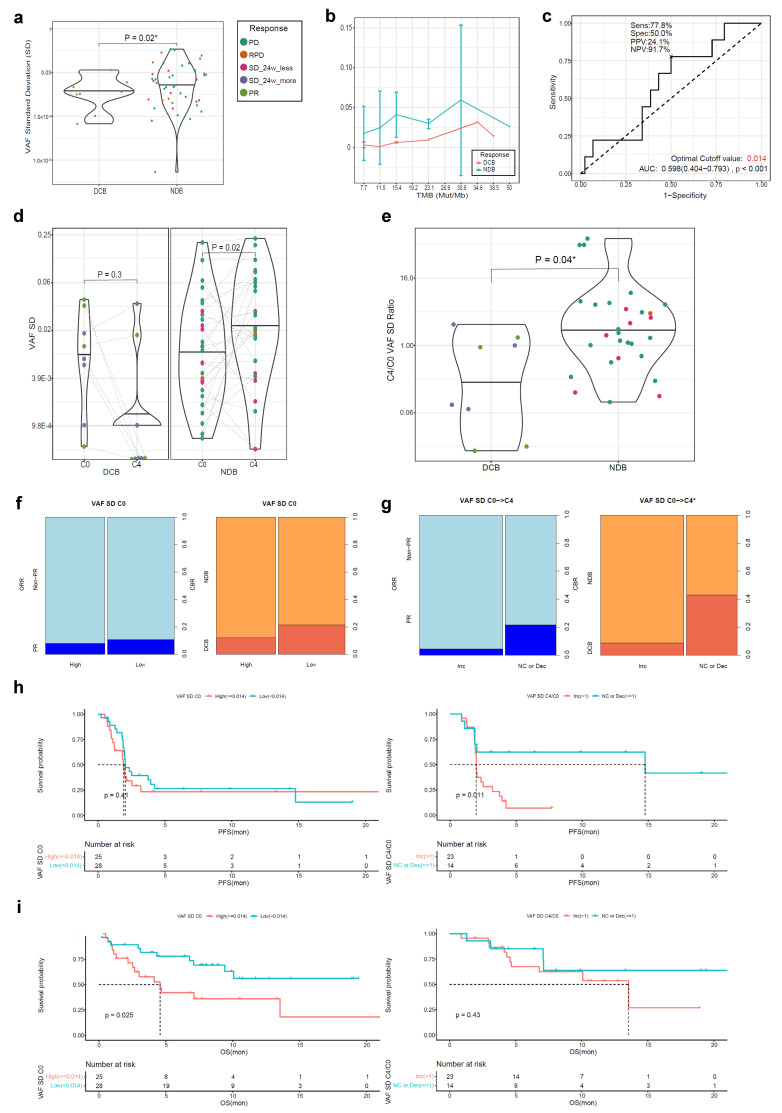
Atezolizumab efficacy according to the VAFSD in the ctDNA BEP (*n* = 53). (**a**) The VAF SD at C0 according to the CBR; (**b**) Comparison between the VAFSD and bTMB at C0; (**c**) The ROC curve for the VAFSD at C0; (**d**) The VAFSD change between C0 and C4/EOT according to the CBR; (**e**) Comparison of the C4/C0 VAFSD ratio according to the CBR; (**f**) Comparison of the ORR and CBR between the C0 VAFSD high and low subgroups; (**g**) Comparison of the ORR and CBR between the subgroups according to the VAFSD dynamics; (**h**) The PFS according to the C0 VAFSD and VAFSD dynamics; (**i**) The OS according to the C0 VAFSD and VAFSD dynamics. * *p* < 0.05. VAFSD, standard deviation of variant allele frequency; ctDNA, circulating tumor DNA; BEP, biomarker-evaluable population; CRB, clinical benefit ratio; circulating cell-free DNA; DCB, durable clinical benefit; NDB, non-durable benefit; PD, progressive disease; RPD, rapid progressive disease; SD, stable disease; PR, partial response; bTMB, blood tumor mutation burden; ROC, receiver operating characteristic; AUC, area under the curve; PPV, positive predictive value; NPV, negative predictive value; ORR, objective response rate; Inc, increased; NC, no change; Dec, decreased; PFS, progression-free survival; OS, overall survival.

**Figure 6 cells-12-01246-f006:**
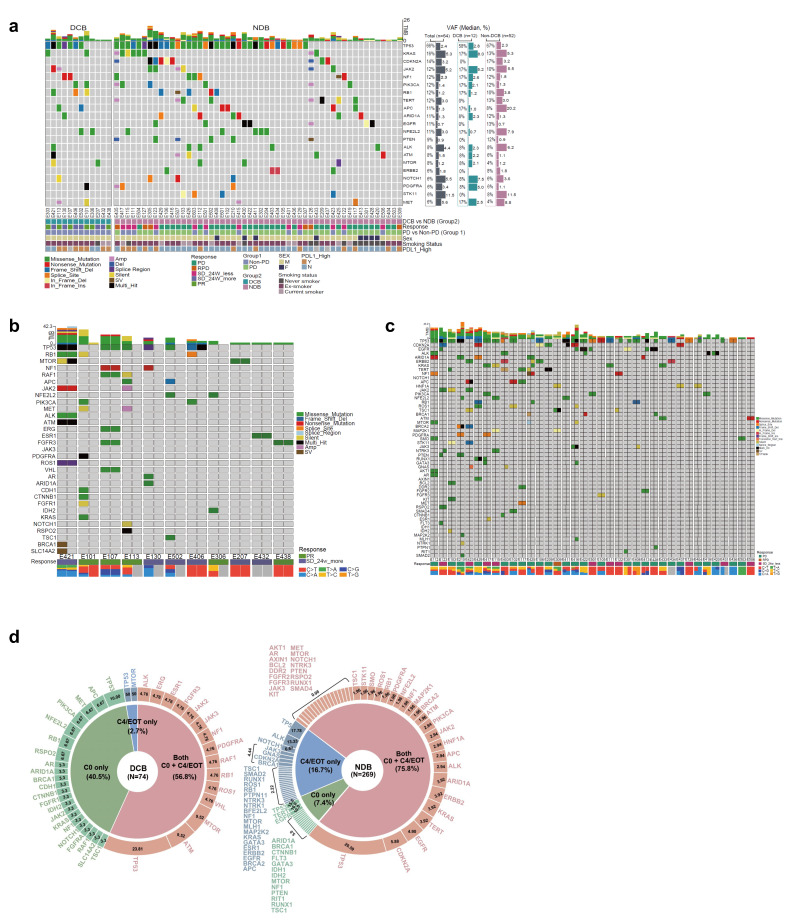
Mutation profiling according to the clinical benefit rate (CBR). (**a**) Overall samples at C0; (**b**) DCB at C0; (**c**) NDB at C0; (**d**) Mutational change between C0 and C4 according to the CBR (DCB and NDB). DCB, durable clinical benefit; NDB, non-durable benefit; TMB, tumor mutation burden; VAF, variant allele frequency; PDL1, programmed death ligand 1; PD, progressive disease; RPD, rapid progressive disease; SD, stable disease; PR, partial response.

**Table 1 cells-12-01246-t001:** Baseline characteristics of the enrolled patients.

Characteristics	No. of Patients, *n* = 100
Age, years	65.0 (42–82)
Sex: Male/Female	84/16
ECOG PS score: 0/1/2	7/92/1
Smoking: Never/Ex/Current smoker	21/66/13
Pack-years	40.0 (1–100)
Histology	
Squamous cell carcinoma	39
Adenocarcinoma	52
NSCLC, NOS	7
Others ^1^	2
Clinical stage (TNM 8th)	
IIIB/IIIC	5/3
IVA/IVB	51/39
Recurrence	2
EGFR mutation, tested	80 (100.0)
Positive	11 (13.8)
Wild type	69 (86.3)
ALK fusion, tested: FISH/IHC/both	74 (100.0): 35/36/3
Positive	1 (1.4)
Negative	73 (98.6)
ROS1 fusion, tested: FISH/PCR/IHC	35 (100.0): 6/28/1
Positive	0 (0.0)
Negative	35 (100.0)
PD-L1 TPS, tested: 22C3/SP263/SP142	95: 67/84/15
22C3: 0%/1–49%/≥50%	24 (35.8)/17 (25.4)/26 (38.8)
SP263: 0%/1–49%/≥50%	27 (32.1)/28 (33.4)/29 (34.5)
SP142 (TC): 0%/1–4%/5–49%/≥50%	11 (73.3)/2 (13.4)/1 (6.7)/1 (6.7)
SP142 (IC): 0%/1–4%/5–9%/≥10%	12 (80.0)/3 (20.0)/0 (0.0)/0 (0.0)
Prior chemotherapy (last)	100
Numbers: 1/2/3/4	67/22/7/4
Aim: Palliative/CCRT ^2^	93/7
Regimen	
Platinum doublet	93
Cytotoxic monotherapy	3
Tyrosine kinase inhibitor	3
Others ^3^	1
Treatment duration, months	5.3 (0.7–36.3)
Best response: CR/PR/SD/PD/Unknown	0/31/44/24/1
Reason for cessation of treatment	
Completion	8
Progression	90
Patient refusal	1
Investigator’s judgment	1

Values are presented as the median (range) or number (%). ^1^ sarcomatoid carcinoma (*n* = 1), NUT carcinoma (*n* = 1). ^2^ neoadjuvant CCRT (*n* = 2), adjuvant CCRT (*n* = 1), definitive CCRT (*n* = 4). ^3^ gemcitabine plus vinorelbine (*n* = 1). ECOG PS, Eastern Cooperative Oncology Group Performance Status; NSCLC, non-small cell lung cancer; NOS, not otherwise specified; EGFR, epidermal growth factor receptor; ALK, anaplastic lymphoma kinase; ROS1, ROS proto-oncogene 1; FISH, fluorescence in situ hybridization; IHC, immunohistochemistry; PCR, polymerase chain reaction; PD-L1, programmed death-ligand 1; TPS, tumor proportional score; TC, tumor cell; IC, immune cell; CCRT, concurrent chemoradiotherapy; CR, complete response; PR, partial response; SD, stable disease; PD, progressive disease.

**Table 2 cells-12-01246-t002:** Efficacy of the atezolizumb treatment in the ITT population.

Variables	No. of Patients, *n* = 100
Atezolizumab cycles	3 (1–32)
Reason for cessation of treatment	
Progression	64
Adverse events	9
Withdrawal of consent	6
Others	21
Best response	
CR	0
PR	10
SD: ≥24weeks/<24 weeks	15/18
PD ^1^	54
NE	3
Objective response rate, %	10
Durable clinical benefit, %	25
Progression	
Confirmed progression	76
No progression	20
Unknown	4
Follow-up duration, months, median (95% CI)	12.3 (10.0–14.6)
PFS, months, median (95% CI)	2.1 (1.6–3.0)
DoR, months, median (95% CI)	NR
OS, months, median (95% CI)	13.1 (10.1–16.2)
Alive	42
Death	39
Lost to follow-up	13
Withdrawal of consent	6

Values are presented as the median (range) or number. ^1^ Patients who showed rapid progression and died before the first assessment of tumor responsiveness were included (*n* = 9). ITT, intention-to-treat; CR, complete response; PR, partial response; SD, stable disease; PD, progressive disease; NE, not evaluable; CI, confidence interval; PFS, progression-free survival; DoR, duration of response; NR, not reached; OS, overall survival.

## Data Availability

The datasets used or analyzed during the current study are available from the corresponding author upon reasonable request.

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
