# Peer review of "Evaluation of Blood Tumor Mutation Burden for the Efficacy of Second-Line Atezolizumab Treatment in Non-Small Cell Lung Cancer: BUDDY Trial"

_cells, 2023, doi:10.3390/cells12091246_

Round 1

Reviewer 1 Report

The present manuscript reports on a retrospective study evaluating the predictive role of baseline bTMB and its dynamic changes during 2nd line atezolizumab in advanced NSCLC. This is a very interesting topic, as the role of bTMB as predictive biomarker for immunotherapy is controversial and conflicting results have been presented to date.

Some minor comments:

·      In the BF1RST trial, patients with a MSAF < 1% had higher ORR with 1st line atezolizumab as compared with those with a MSAF ≥ 1% (Kim ES, et al. Nat Med 2022). This phenomenon might be attributable to the lower tumor burden of these patients, a well-known clinical parameter associated with better immunotherapy efficacy. This data should be discussed.

·      Recently, the preliminary results of the prospective study BFAST (cohort C) (Peters S, et al. Nat Med 2022) were published. Using a cut-off of bTMB of 16 (FoundationOne CDx Liquid platform), they showed that atezolizumab failed to improve PFS as compared with platinum-based chemotherapy in high bTMB patients. The results of this study should be discussed here.

Author Response

Manuscript ID: cells-2308753

Title: Evaluation of Blood Tumor Mutation Burden for the Efficacy of Second-Line Atezolizumab Treatment in Non-Small Cell Lung Cancer: BUDDY Trial

Dear Editors of Cells,

Thank you for giving us the opportunity to submit a revised manuscript again.
We appreciate the time and effort that you and the reviewers dedicated to providing feedback on our manuscript and are grateful for the insightful comments to improve our paper.
We have incorporated an additional suggestion made by the reviewers.
This change is highlighted in yellow as a point-by-point response to the reviewer’s comment. The revisions by the professional English editor, Charles A. are marked up using the “Track Changes” function using MS Word. And the reference formatting was also revised according to the instruction for the authors.

We hope that all changes we have made meet with your approval and look forward to your response.

Sincerely yours,

Sung-Min Chun and In-Jae Oh, on behalf of all the authors.

*Corresponding author: Sung-Min Chun, PhD

Department of Pathology, Asan Medical Center,

88 Olympic-ro 43 Gil, Songpa-gu, Seoul 05505, Republic of Korea

Tel.: 02-3010-5999, Fax: 02-472-7898

*Corresponding author: In-Jae Oh, MD, PhD

Department of Internal Medicine, Chonnam National University Hwasun Hospital,

322 Seoyang-ro, Hwasun, Jeonnam 58128, Republic of Korea

Tel.: 061-379-7617, Fax: 061-379-7619

[Reviewer 1`s comments]
The present manuscript reports on a retrospective study evaluating the predictive role of baseline bTMB and its dynamic changes during 2nd line atezolizumab in advanced NSCLC. This is a very interesting topic, as the role of bTMB as predictive biomarker for immunotherapy is controversial and conflicting results have been presented to date.

Some minor comments:

Comment 1: In the BF1RST trial, patients with a MSAF < 1% had higher ORR with 1st line atezolizumab as compared with those with a MSAF ≥ 1% (Kim ES, et al. Nat Med 2022). This phenomenon might be attributable to the lower tumor burden of these patients, a well-known clinical parameter associated with better immunotherapy efficacy. This data should be discussed.

Reply 1: Thank you for your valuable comments. In the BF1RST trial, bTMB could not be applied for biomarker analysis because the MSAF <1% population was non-evaluable for bTMB. In comparisons of baseline characteristics between MSAF <1% and MSAF ³1% subgroups, the MSAF <1% population had more favorable factors, such as lower age, fewer current smokers, more patients with PD-L1 positive status, lower number of target lesions, and smaller tumor size, which may suggest a lower tumor burden in MSAF <1% population. Although the difference in prognostic factors turned out to be insignificant after propensity score analysis, the MSAF might be a potentially independent biomarker reflecting the tumor burden. We added this discussion as you commented and highlighted them in yellow.

Changes in the text: (Page 15, Line 457-464) In the B-F1RST trial, ~.   

Comment 2: Recently, the preliminary results of the prospective study BFAST (cohort C) (Peters S, et al. Nat Med 2022) were published. Using a cut-off of bTMB of ≥16 (FoundationOne CDx Liquid platform), they showed that atezolizumab failed to improve PFS as compared with platinum-based chemotherapy in high bTMB patients. The results of this study should be discussed here.

Reply 2: We added the reference in the discussion section as you commented.  

Changes in the text: (Page 15, Line 422-425) In the primary analysis of the BFAST ~.

Reviewer 2 Report

The study proposes the potential of using blood-based biomarkers to predict treatment outcomes in patients with NSCLC and explores the potential of bTMB and other blood-based biomarkers in predicting response to immune checkpoint inhibitors. The study results show that higher bTMB levels are associated with higher objective response rates (ORR) and longer progression-free survival (PFS). Additionally, the article introduces other blood-based biomarkers using circulating tumor DNA and explores their potential in predicting response to immune checkpoint inhibitors.

Overall,if the issues I raised are addressed, this research work will be greatly improved and ready for publication:

1. Please verify the units of the vertical coordinate TMB in Figure 2a.

2. Please ensure that the note below the picture (DCR) matches the note in the picture (CBR) in Figure 2fg, Figure 3ef, Figure 4fg, and Figure 5fg.

3. In Figure 3, please correct the picture number in the comment below the picture by adding an "f" and removing an "i".

4. Please ensure that the effective digits of percent on Figure 6d are consistent with those in Text .

5. The study only includes a small number of patients and only uses one type of immune checkpoint inhibitor for treatment. Therefore, larger clinical trials are needed to validate these results and determine the effectiveness of these blood-based biomarkers in different types of tumors and treatment regimens.

6. The article needs to delve deeper into the relationship between these blood-based biomarkers and other clinical and molecular features to better understand their role in tumor development and treatment.

7. In the article, there are some inappropriate word choices, grammar errors, and spelling mistakes that need to be corrected. I suggest that the authors carefully check their language expression during revision to ensure accuracy and clarity.

Based on these issues, I recommend that the authors revise and improve their article. If these issues are addressed, this study will provide stronger evidence for using blood-based biomarkers to predict treatment outcomes in patients with NSCLC and guide future clinical trials and treatment regimens.

Author Response

Manuscript ID: cells-2308753

Title: Evaluation of Blood Tumor Mutation Burden for the Efficacy of Second-Line Atezolizumab Treatment in Non-Small Cell Lung Cancer: BUDDY Trial

Dear Editors of Cells,

Thank you for giving us the opportunity to submit a revised manuscript again.
We appreciate the time and effort that you and the reviewers dedicated to providing feedback on our manuscript and are grateful for the insightful comments to improve our paper.
We have incorporated an additional suggestion made by the reviewers.
This change is highlighted in yellow as a point-by-point response to the reviewer’s comment. The revisions by the professional English editor, Charles A. are marked up using the “Track Changes” function using MS Word. And the reference formatting was also revised according to the instruction for the authors.

We hope that all changes we have made meet with your approval and look forward to your response.

Sincerely yours,

Sung-Min Chun and In-Jae Oh, on behalf of all the authors.

*Corresponding author: Sung-Min Chun, PhD

Department of Pathology, Asan Medical Center,

88 Olympic-ro 43 Gil, Songpa-gu, Seoul 05505, Republic of Korea

Tel.: 02-3010-5999, Fax: 02-472-7898

*Corresponding author: In-Jae Oh, MD, PhD

Department of Internal Medicine, Chonnam National University Hwasun Hospital,

322 Seoyang-ro, Hwasun, Jeonnam 58128, Republic of Korea

Tel.: 061-379-7617, Fax: 061-379-7619

 [Reviewer 2`s comments]
The study proposes the potential of using blood-based biomarkers to predict treatment outcomes in patients with NSCLC and explores the potential of bTMB and other blood-based biomarkers in predicting response to immune checkpoint inhibitors. The study results show that higher bTMB levels are associated with higher objective response rates (ORR) and longer progression-free survival (PFS). Additionally, the article introduces other blood-based biomarkers using circulating tumor DNA and explores their potential in predicting response to immune checkpoint inhibitors.

Overall, if the issues I raised are addressed, this research work will be greatly improved and ready for publication:

Based on these issues, I recommend that the authors revise and improve their article. If these issues are addressed, this study will provide stronger evidence for using blood-based biomarkers to predict treatment outcomes in patients with NSCLC and guide future clinical trials and treatment regimens.

Comment 1: Please verify the units of the vertical coordinate TMB in Figure 2a.

Reply 1: The unit of TMB is mutations per megabase (Mut/Mb). We revised the TMB unit in Figure 2a as Mut/Mb consistent with those in the other figures and text and highlighted them in yellow.  

Changes in the text: (Page 8, figure 2a) TMB (Muts/Mb) à TMB (Mut/Mb); (Page 7, line 240, 246 & Page 9, line 307) mut/Mb à Mut/Mb

Comment 2: Please ensure that the note below the picture (DCR) matches the note in the picture (CBR) in Figure 2fg, Figure 3ef, Figure 4fg, and Figure 5fg.

Reply 2: We switched “DCR” to “CBR” in the comments below figures consistent with those in the figures and text and highlighted them in yellow. 

Changes in the text: (Page 8, line 260, 261; Page 10, line 320, 321; Page 11, line 335; Page 13, line 392) DCR à CBR

Comment 3: In Figure 3, please correct the picture number in the comment below the picture by adding an "f" and removing an "i".

Reply 3: We revised the numbers of the comments below Figure 3f-h as you indicated and highlighted them in yellow.    

Changes in the text: (Page 10, line 320-322)  

Comment 4: Please ensure that the effective digits of percent on Figure 6d are consistent with those in Text.

Reply 4: We modified the percentages (incidence) of variants in the DCB group as one digit below the decimal point and revised the numbers in Figure 6d consistent with those in the text.   

Changes in the text: (Page 12, line 364, 366) 40.5%, 2.7%; (Page 14, Figure 6d) C0 only (40.5%), C4/EOT (2.7%), Both C0 + C4/EOT (56.8%)   

Comment 5: The study only includes a small number of patients and only uses one type of immune checkpoint inhibitor for treatment. Therefore, larger clinical trials are needed to validate these results and determine the effectiveness of these blood-based biomarkers in different types of tumors and treatment regimens.

Reply 5: We added the comments for the necessity of further trials including different types of tumors and immune checkpoint inhibitors in order to validate the utility of blood-based biomarkers in the discussion section.     

Changes in the text: (Page 16, line 503-505) Furthermore, larger clinical trials ~.

Comment 6: The article needs to delve deeper into the relationship between these blood-based biomarkers and other clinical and molecular features to better understand their role in tumor development and treatment.

Reply 6: We described the necessity of comprehensive molecular analysis for plasma matched with tumor tissue of individual patients in the discussion section. We are trying to establish a solid plan for the subsequent cohort study to validate of the biomarkers.   

Changes in the text: (Page 16, line 502-503) A comprehensive mutational profiling ~.

Comment 7: In the article, there are some inappropriate word choices, grammar errors, and spelling mistakes that need to be corrected. I suggest that the authors carefully check their language expression during revision to ensure accuracy and clarity.

Reply 7: We already had an English editing service before submitting this article. In addition, we double-checked and revised several minor errors during revising this manuscript. The revisions by the professional English editor are marked up using the “Track Changes” function using MS Word. Please find the attached certificate.

Round 2

Reviewer 2 Report

Based on my review of the authors' response and revisions to the manuscript, I have decided to accept this article. I appreciate the authors' efforts in addressing my concerns and making necessary improvements to the manuscript.

Author Response

Comment 1: Based on my review of the authors' responses and revisions to the manuscript, I have decided to accept this article. I appreciate the authors' efforts in addressing my concerns and making necessary improvements to the manuscript.

Reply 1: Thank you for your feedback and effort in reviewing my paper.